# PeerJ

# Differential genotoxicity of diphenyl diselenide (PhSe)$_2$ and diphenyl ditelluride (PhTe)$_2$

Daiane Francine Meinerz, Josiane Allebrandt, Douglas O.C. Mariano, Emily P. Waczuk, Felix Antunes Soares, Waseem Hassan and João Batista T. Rocha

Departamento de Bioquímica e Biologia Molecular, Centro de Ciências Naturais e Exatas, Universidade Federal de Santa Maria, Santa Maria, RS, Brasil

## ABSTRACT

Organoselenium compounds have been pointed out as therapeutic agents. In contrast, the potential therapeutic aspects of tellurides have not yet been demonstrated. The present study evaluated the comparative toxicological effects of diphenyl diselenide (PhSe)$_2$ and diphenyl ditelluride (PhTe)$_2$ in mice after *in vivo* administration. Genotoxicity (as determined by comet assay) and mutagenicicity were used as end-points of toxicity. Subcutaneous administration of high doses of (PhSe)$_2$ or (PhTe)$_2$ (500 µmol/kg) caused distinct genotoxicity in mice. (PhSe)$_2$ significantly decreased the DNA damage index after 48 and 96 h of its injection ($p < 0.05$). In contrast, (PhTe) caused a significant increase in DNA damage ($p < 0.05$) after 48 and 96 h of intoxication. (PhSe)$_2$ did not cause mutagenicity but (PhTe)$_2$ increased the micronuclei frequency, indicating its mutagenic potential. The present study demonstrated that acute *in vivo* exposure to ditelluride caused genotoxicity in mice, which may be associated with pro-oxidant effects of diphenyl ditelluride. In addition, the use of this compound and possibly other related tellurides must be carefully controlled.

## INTRODUCTION

Selenium (Se) and Tellurium (Te) belongs to the chalcogen family, sharing similar electronic configuration and some chemical properties with sulfur (S) (*Comasseto et al., 1997*; *Comasseto, 2010*). Se has a fundamental role in several living organisms as component of several antioxidant enzymes, including glutathione peroxidase and thioredoxin reductase (*Arner & Holmgren, 2000*; *Nogueira & Rocha, 2011*). Despite its biological role, the excess of selenium can be toxic due its ability to generate free radicals and catalyze thiol oxidation (*Barbosa et al., 1998*; *Nogueira, Zen & Rocha, 2004*; *Rocha et al., 2012*; *Hassan & Rocha, 2012*; *Kade, Balogun & Rocha, 2013*). The excess of free radical formation can damage mammalian tissues including thiol containing enzymes that are sensitive to pro-oxidant situations (*Rocha et al., 2012*; *Rosa et al., 2007*; *Maciel et al., 2000*). Diphenyl diselenide (PhSe)$_2$, (Fig. 1) is a simple and stable organoselenium compound

Corresponding authors
Waseem Hassan,
waseem_anw@yahoo.com
João Batista T. Rocha,
jbtrocha@yahoo.com.br

**Figure 1 Structure of diphenyl diselenide and diphenyl ditelluride.**

widely used in organic synthesis and it has been proposed as a good candidate for pharmacological and therapeutic purposes (*Nogueira, Zen & Rocha, 2004*; *Rosa et al., 2007*; *Nogueira & Rocha, 2011*). (PhSe)$_2$ exhibits thiol peroxidase-like activity superior to that of ebselen, an organoselenium compound that has been used in clinical trials as antioxidant and mimetic of native glutathione peroxidase enzymes (*Nogueira & Rocha, 2011*; *Kade & da Rocha, 2013*; *Kade, Balogun & Rocha, 2013*). However, exposure to high doses of (PhSe)$_2$ can deplete thiols in different tissues and can be neurotoxic to rodents (*Maciel et al., 2000*). The LD50 of diphenyl diselenide is 210 µmol/kg (intraperitoneal) or greater than 500 µmol/kg (subcutaneous) in adult mice (*Nogueira et al., 2003*).

There are reports that trace amounts of Te are present in body fluids such as blood and urine (*Chasteen et al., 2009*). Te has also been found in the form of tellurocysteine and telluromethionine in several proteins in bacteria, yeast and fungi but telluroproteins have not been identified in animal cells (*Bienert, Schussler & Jahn, 2008*). Thus, in contrast to selenium, tellurium does not have physiological functions (*Taylor, 1996*). Literature has demonstrated immunomodulatory, antioxidant and anticancer properties of various organotellurides (*Nogueira, Zen & Rocha, 2004*; *Avila et al., 2012*), semisynthetic telluro-subtilisin (*Mao et al., 2005*) and dendrimeric organotellurides (*Francavilla et al., 2001*). More sophisticated telluride molecules were synthesized from polystyrene nanoparticle via microemulsion polymerization. The nanoenzyme showed higher efficiency and provided a platform for the synthesis and designing of polymeric nanoparticles as excellent model of enzyme mimics (*Huang et al., 2008*). Organotellurium compounds can also mimic glutathione peroxidase activity (*Engman et al., 1995*) and, consequently, these compounds can be potential antioxidants, effective against hydrogen peroxide, peroxynitrite, hydroxyl radicals and superoxide anions (*Andersson et al., 1994*; *Kanski et al., 2001*; *Jacob et al., 2000*).

Recently, our research group demonstrated that organoselenium and organotellurium present hemolytic and genotoxic effects in human blood cells (*Santos et al., 2009a*; *Santos et al., 2009b*; *Caeran Bueno et al., 2013*), which is in accordance with results published by other laboratories in experimental bacteria and rodent models (*Degrandi et al., 2010*). Similarly, organoselenides and tellurides can be toxic in different *in vivo* and *in vitro* models of animal pathologies (*Maciel et al., 2000*; *Taylor, 1996*; *Stangherlin, Rocha & Nogueira, 2009*; *Moretto et al., 2007*; *Heimfarth et al., 2011*; *Heimfarth et al., 2012a*; *Heimfarth et al., 2012b*; *Comparsi et al., 2012*). In effect, diphenyl ditelluride (PhTe)$_2$ was found to be extremely toxic to mice and rats after acute or chronic exposure (*Maciel et al., 2000*; *Heimfarth et al., 2012b*; *Comparsi et al., 2012*). The toxicity of tellurides can be associated with their pro-oxidant activity, particularly, the oxidation of thiol- and selenol-groups of proteins (*Nogueira, Zen & Rocha, 2004*; *Comparsi et al., 2012*; *Hassan & Rocha, 2012*).

Following our interest to determine the boundary between the potential protective and toxic properties of organochalcogens, the present study was designed to evaluate the toxic potential of $(PhSe)_2$ and $(PhTe)_2$ in mice. We have determined the genotoxicity and mutagenicity of these compounds after acute administration to Swiss male mice, using DNA damage and micronuclei frequency as end-points of toxicity.

## MATERIAL AND METHODS

### Chemicals

The chemical structure of organochalcogens tested in this study is shown in Fig. 1 diphenyl diselenide and diphenyl ditelluride. The compounds were dissolved in canola oil immediately before use. $(PhSe)_2$ and $(PhTe)_2$ were obtained from Sigma-Aldrich. All other chemicals were of analytical grade and obtained from standard commercial suppliers.

### Animals

Male Swiss adult mice weighing 30–40 g were obtained from our own breeding colony (Animal house-holding, UFSM-Brazil). Animals were kept in separate animal cages, on a 12-h light/dark cycle, at a room temperature of $(23 \, °C \pm 3)$ and with free access to food and water. The animals were used according to the guidelines of the committee on care and use of experimental animal resources of the Federal University Of Santa Maria, Brazil (23081.002435/2007-16).

Mice were divided in six groups ($n = 5$) and received one subcutaneous injection of (1) canola oil (Control group 48 h, mice were euthanized 48 h after the oil injection); (2) diphenyl ditelluride (500 µmol/kg in canola oil, euthanized 48 h after injection) ; (3) diphenyl diselenide (500 µmol/kg in canola oil, euthanized 48 h after injection); (4) canola oil (Control group 96 h, mice were euthanized 96 h after injection); (5) diphenyl ditelluride (500 µmol/kg in canola oil, euthanized 96 h after injection) and (6) diphenyl diselenide (500 µmol/kg in canola oil, euthanized 96 h after injection). The doses were based in a previous acute toxicological study by *Maciel et al. (2000)*.

### Sample preparation for comet assay

Mice were anesthesized with ketamine and 2.5 ml blood samples were collected by heart puncture and immediately euthanized by decaptation. Mice blood leukocytes were isolated and used in the comet test but no pre-incubation was carried out (*Santos et al., 2009a*; *Santos et al., 2009b*; *Meinerz et al., 2011*).

### Micronucleus test

In a micronucleus test (MN), two samples of blood from each animal were placed in a microscope slides and air dried at room temperature. Slides were stained with 5% May-Grunwald-Giemsa for 5 min. The criteria used for the identification of MN were a size smaller than one-third of the main nucleus, no attachment to the main nucleus, and identical color and intensity as in the main nucleus. MN were counted in 2000 cells

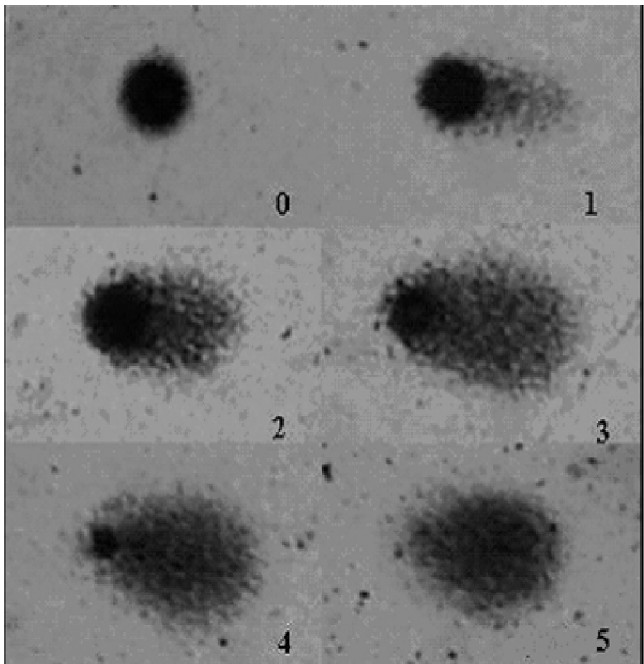

**Figure 2 DNA damage quantification.** Classifications of DNA damage in human leukocytes. DNA damage index was calculated from cells in different damage levels, which were classified in the visual score by the measurement of DNA migration length and in the amount of DNA in the tail. The level 5 was excluded from our evaluation.

with well-preserved cytoplasm and calculated as: % MN = number of cells containing micronucleus × 100/total number of cells counted. Micronuclei presence was determined by three investigators that were blind to the animal treatments.

## Comet assay

Comet assay is a rapid, simple and sensitive technique for measuring DNA breaks in single cells. This test has been used to investigate the effect of many toxic agents on DNA (*Collins & Harrington, 2002*; *Blasiak, Arabski & Krupa, 2004*). The comet assay was performed under alkaline conditions according to the procedures described by *Santos et al. (2009a)* and *Santos et al. (2009b)*. The slides obtained from white blood cells of treated mice were analyzed under blind conditions by at least two individuals. DNA damage is presented as DNA damage index (DI). The DNA damage was calculated from cells in different damage classes (completely undamaged: 100 cells × 0 to maximum damaged −100 cells × 4). Damage index is illustrated in Fig. 2 and classes were determined considering the DNA tail and DNA migration length.

## Statistical analysis

Data are expressed as mean ± SD from five independent experiments performed in duplicate or triplicate. Statistical analysis was performed using a Kruskal-Wallis Test followed by Dun's test. Results were considered statistically significant when $p < 0.05$.

Table 1 DNA damage levels in leukocytes from mice treated with diselenide or ditelluride.

| Compound | Hours of exposition | Damage levels of DNA | | | | | DI |
|---|---|---|---|---|---|---|---|
| | | **0** | **1** | **2** | **3** | **4** | |
| **Control** | **48 h** | $61.0 \pm 0.5$ | $19.6 \pm 2.0$ | $13.4 \pm 1.4$ | $4.5 \pm 0.8$ | $1.0 \pm 0.5$ | $63.0 \pm 2.5$[a] |
| **(PhSe)$_2$** | **48 h** | $77.2 \pm 3.6$ | $11.8 \pm 1.6$ | $6.6 \pm 1.3$ | $3.8 \pm 1.1$ | $0.6 \pm 0.2$ | $40.8 \pm 7.8$[b] |
| **(PhTe)$_2$** | **48 h** | $48.0 \pm 9.7$ | $32.3 \pm 9.6$ | $13.0 \pm 3.2$ | $5.0 \pm 1.0$ | $1.6 \pm 0.6$ | $80.0 \pm 9.3$[c] |
| **Control** | **96 h** | $63.5 \pm 0.5$ | $20.7 \pm 6.5$ | $12.5 \pm 5.5$ | $3.7 \pm 0.5$ | $0.0 \pm 0.0$ | $58.0 \pm 4.6$[a] |
| **(PhSe)$_2$** | **96 h** | $80.0 \pm 2.0$ | $10.0 \pm 2.0$ | $5.0 \pm 3.0$ | $3.0 \pm 0.6$ | $2.0 \pm 2.0$ | $40.0 \pm 1.1$[b] |
| **(PhTe)$_2$** | **96 h** | $59.5 \pm 3.5$ | $19.0 \pm 7.0$ | $12.0 \pm 3.0$ | $9.2 \pm 0.8$ | $1.6 \pm 0.5$ | $76.0 \pm 1.2$[c] |

**Notes.**

Distribution of damage levels in mice leukocytes exposed to diphenyl diselenide and diphenyl ditelluride (500 µmol/kg, s.c.). DNA damage is presented as DNA damage index (DI). Data are expressed as means for five independent experiments. Statistical analysis by a Kruskal-Wallis Test test followed by Dun's test.

## RESULTS

No animal died during the experimental period. After 48 h of diselenide or ditelluride treatment, mice did not show symptoms of toxicity such as stereotypical behavior, ataxia, diarrhea, increased dieresis or abdominal writings. However, after 96 h, the group treated with (PhTe)$_2$ presented diarrhea, low level of motor activity and a decrease in body weight (data not shown); which is in accordance with previous finding from our laboratory (*Maciel et al., 2000*).

### Comet assay

After *in vivo* administration, diphenyl diselenide caused a significant decrease in DNA damage index (DI) both after 48 and 96 h. In contrast, diphenyl ditelluride caused a significant increase in DNA damage index (DI). After 48 h, the damage caused by ditelluride was about 25 and 100% higher than control and diphenyl diselenide groups, respectively (Table 1). After 96 h, the DI caused by diphenyl ditelluride was about 30 and 90% higher than control and diselenide treated mice, respectively (Table 1).

### Micronucleus test

After 48 or 96 h of a single dose of diphenyl ditelluride, there was a significant increase in the number of micronuclei in mice when compared with control and diphenyl diselenide group (Fig. 3). Diphenyl diselenide did not modify the number of micronuclei when compared to the control group (Fig. 3).

## DISCUSSION

The selected dose of both chalcogens was based on our previous report (*Maciel et al., 2000*), where we tested different doses for acute and chronic exposure. Similarly, in the same dose range, diphenyl diselenide has been reported to have interesting pharmacological effects, such as antinoception and anti-inflammatory effects, among others, (see, for instance, *Savegnago et al., 2008*; *Savegnago et al., 2007a*; *Savegnago et al., 2007b* and *Savegnago et al., 2006*). However, it must be emphasized here that in this range of doses, it also causes

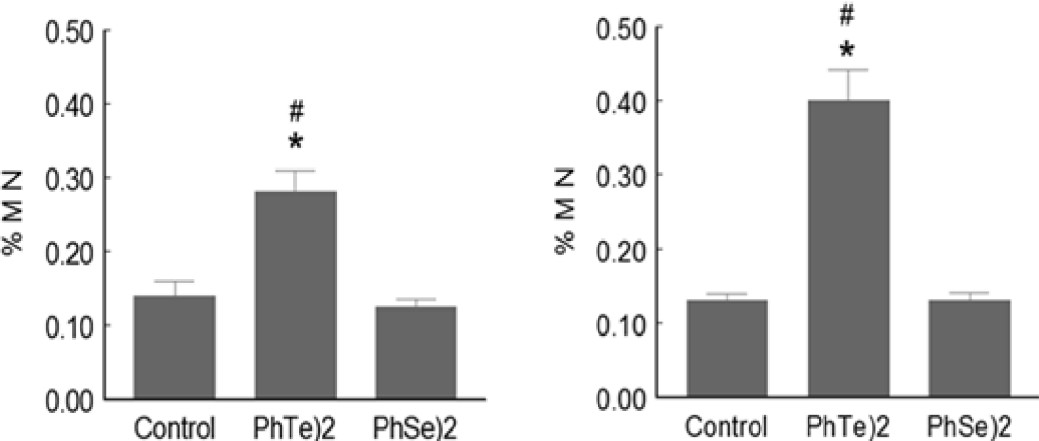

**Figure 3 Micronuclei frequency after treatment with diselenide and ditelluride.** Frequency of Micronuclei (MN) cells in mice exposed to (PhTe)$_2$ or (PhSe)$_2$. Mice were exposed to a single dose of diselenide or ditelluride (500 μmol/kg, s.c.). Forty eight and 96 h after the injection, blood cells were examined for the presence of micronuclei. Data are expressed as mean ± SD for 5 mice per group. ∗ denoted $p > 0.01$ as compared to control group; # Denoted $p > 0.01$ as compared to diphenyl diselenide.

toxicity in mice and rats (*Nogueira et al., 2003*; *Nogueira & Rocha, 2011*). Consequently, the acute use of diphenyl diselenide may be possible, but its chronic or repeated use is unfeasible.

The results presented here indicate clear toxic effects of (PhTe)$_2$ when compared with (PhSe)$_2$. Tellurium (Te) has the potential of redox cycling which leads to formation of reactive oxygen species (ROS) which can damage biomolecules (*Maciel et al., 2000*; *Nogueira, Zen & Rocha, 2004*; *Santos et al., 2009a*; *Santos et al., 2009b*; *Degrandi et al., 2010*; *Sailer et al., 2004*; *Caeran Bueno et al., 2013*). Organotellurium-induced intracellular ROS accumulation has been reported to be the cause of cell death in HL-60 and different types of cancer cells (*McNaughton et al., 2004*; *Sandoval et al., 2010*; *Ding et al., 2002*; *Rigobello et al., 2009*). In contrast, exposure of mice to (PhSe)$_2$ caused a significant decrease in the DNA damage index (DI) both after 48 and 96 h of drug administration as shown in Table 1. The protective effect can be attributed to its antioxidant or GPx like activity (*Nogueira & Rocha, 2011*).

As observed in DNA damage test, the toxic behavior of (PhTe)$_2$ was completely different than (PhSe)$_2$ in micronucleus assay. The frequency of mutations, showed by an increase of micronuclei frequency, reinforce the toxicity of (PhTe)$_2$. It is important to note that (PhSe)$_2$ did not modify the number of micronuclei, when compared to the control group (Fig. 3). Previous studies have also demonstrated mutagenicicity of (PhTe)$_2$ at higher concentrations in V79 cells (*Rosa et al., 2007*). We have also

reported the mutagenicity of another Te-containing organic compound, (*S*)-dimethyl 2-(3-(phenyltellanyl) propanamido) succinate in mice leukocytes (*Meinerz et al., 2011*)

In conclusion, the results presented here indicate that diphenyl ditelluride is toxic to mice, whereas at the same dose diphenyl diselenide had protective effects. These effects may be linked to the pro-oxidant activity exhibited by organotellurium compounds. This data supports studies that have been published about the toxicological and pharmacological effects of organochalcogens in different pathological models. In effect, our data indicated that diphenyl diselenide can have protective effects after *in vivo* administration to mice, which can be related to its antioxidant properties, whereas diphenyl ditelluride is much more toxic than diphenyl diselenide. Furthermore, in view of the genotoxic effect of $(PhTe)_2$, the indication in the literature that organotellurides could be therapeutically active compounds must be revisited taking into consideration the potential toxicity of this element. Accordingly, additional studies will be needed to elucidate the mechanism(s) by which $(PhTe)_2$ mediates its toxicity and whether or not distinct chemical forms of organotellurides can have a similar toxic effect in animal models.

### Funding
Financial support was received from TWAS, CAPES, SAUX, PROAP, CNPq, VITAE, CNPq-INCT for Excitotoxicity and Neuroprotection, FAPERGS-PRONEX and FAPERGS. Waseem Hassan and JBTR are recipients of CNPq Fellowships. The funders had no role in study design, data collection and analysis, decision to publish, or preparation of the manuscript.

### Grant Disclosures
The following grant information was disclosed by the authors:
TWAS, CAPES, SAUX, PROAP, CNPq, VITAE, CNPq-INCT.
FAPERGS-PRONEX and FAPERGS.

### Competing Interests
João Batista T. Rocha is an Academic Editor for PeerJ.

### Author Contributions

- Daiane Francine Meinerz performed the experiments, analyzed the data, contributed reagents/materials/analysis tools, wrote the paper, prepared figures and/or tables, reviewed drafts of the paper.
- Josiane Allebrandt performed the experiments, analyzed the data, contributed reagents/materials/analysis tools, wrote the paper, prepared figures and/or tables.
- Douglas O.C. Mariano performed the experiments, contributed reagents/materials/analysis tools, reviewed drafts of the paper.
- Emily P. Waczuk performed the experiments, contributed reagents/materials/analysis tools.

- Felix Antunes Soares conceived and designed the experiments.
- Waseem Hassan conceived and designed the experiments, analyzed the data, contributed reagents/materials/analysis tools, wrote the paper, prepared figures and/or tables, reviewed drafts of the paper.
- João Batista T. Rocha conceived and designed the experiments, analyzed the data, wrote the paper, prepared figures and/or tables, reviewed drafts of the paper.

## Animal Ethics

The following information was supplied relating to ethical approvals (i.e., approving body and any reference numbers):

The guidelines of the committee on care and use of experimental animal resources of the Federal University Of Santa Maria, Brazil (23081.002435/2007-16).

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
