# Peer review of "Differential genotoxicity of diphenyl diselenide (PhSe)2 and diphenyl ditelluride (PhTe)2"

_PeerJ, doi:10.7717/peerj.290_

## Round 0.1 · original submission · Major Revisions

Thank you for your submission. Two reviewers have evaluated the manuscript. While the study has merit, there are several issues to be addressed. One is the dose-response issue, as “dose makes a poison”; there are over a dozen English errors that need to be carefully proof-reading to correct. Please respond to all reviewer comments. You may wish to consult with a language editing service. I agree with reviewer #1 that multiple (at least two) doses are recommended to validate toxic effect of these compounds in a dose-dependent manner and more in-depth studies are desired to provide mechanistic basis of differential toxicity of (PhSe)2 and (PhTe)2. The manuscript should be adjusted according to the comments of the reviewers.

In addition, the Figure 2 legend and Table 1 note are incomplete. The tile of the paper should be changed to address the differential genotoxicity of diphenyl diselenide (PhSe)2 and diphenyl ditelluride (PhTe)2, rather than “protective effects of diphenyl diselenide (PhSe)2”, as no toxicants are used for (Phse)2 to protect against.

Reviewer 1 ·

Basic reporting

In general, this manuscript is technically sound and the conclusion is made from reasonable experimental observations. Several major comments are suggested for improvement.

Experimental design

Reasonable experiments are design for major conclusion. However, more doses of the compounds are suggested to show the dose range for a toxic/protective effect.

Validity of the findings

Proper controls are used, and the conclusion is made from proper analysis of the data.

Additional comments

In the manuscript entitled “Diphelyl Ditelluride Causes Acute Genotoxicity in Adult Mice, Whereas Diphenyl Diselenide Has a Protective Effect”, the authors compared genotoxicity and mutagenicicity using Comet assay and micronuclei frequency assay, respectively. The authors made the conclusion that diphelyl ditelluride causes genotoxicity whereas diphenyl diselenide has a protective effect. In general, this manuscript is technically sound and the conclusion is made from reasonable experimental observations. Several major comments are suggested for improvement.
1. The authors nicely showed results from comet assay and micronuclei frequency assay as a function of time. However, it is equally important to show dose-response of these two compounds on DNA damage and mutagenicicity. Thus, multiple (at least two) doses are recommended to validate toxic or protective effect of these compounds in a dose-dependent manner.
2. The authors heavily discussed the potential mechanism of diphelyl ditelluride caused genotoxicity, and proposed that oxidative stress and ROS may serve as a major mechanism. Thus, assays measuring redox state of the cells are highly recommended (for example, GSH/GSSG ratio, ROS staining).
3. There are some errors in the text in grammar or spelling. Please proof-read again carefully.
4. In method section, if four animals were used in each treatment group, n should be 4 rather than 6.

·

Basic reporting

This is an excellent manuscript from a group that has a long history of studying these compounds, The manuscript is sound in terms of its analysis and methods, extending the understanding of the biological effects and toxicological effects of diphenyl diselenide (PhSe)2 and diphenyl ditelluride (PhTe)2. Specifically, herein the authors addressed issues associated with genotoxicity (as determined by comet Assay) and mutagenicicity as end-points of toxicity in mice exposed to the 2 compounds.

Experimental design

The methods are sound. I only ask the authors to expand a bit and explain their choice for a non-parametric test. There needs to be some discussion on the distribution of the data.

Validity of the findings

The findings are novel and to my knowledge this is a first comparison of toxic endpoints contrasting diphenyl diselenide and diphenyl ditelluride in mice after in vivo administration. The findings suggest that caution must be used with the latter as it does have a genotoxic effect.

Additional comments

Overall, this is a novel paper.
My only suggestion is regarding the statistics.
I also noted few errors in the text, please proof-read again carefully.

---

## Round 0.2 · Minor Revisions

Your revised manuscript has been improved. However, there are still some additional concerns that needs minor reversion.

·

Basic reporting

No comments

Experimental design

No Comments

Validity of the findings

No Comments

Additional comments

This manuscript reports the Differential Genotoxicity of Diphenyl Diselenide (PhSe)2 and Diphenyl Ditelluride (PhTe)2.
The methods are sound, and the results are interpreted reasonably.

Reviewer 2 ·

Basic reporting

The manuscript entitled "Differential Genotoxicity of Diphenyl Diselenide (PhSe)2 and Diphenyl Ditelluride (PhTe)2" authored by Meinerz et al desxribes further mechanistic effects of diphenyl diselenide and ditelluride in mice. It is already known that these compounds have toxic and pharmacological properties, that thay cause oxidative stress, lipid peroxidation, and other effects, but DNA effects have never been demonstrated. Authors investigate findings from an article published by Maciel et al in 2000. Methdology and results are simple, but very significant when considering that the group seeks for the safe use of these compounds as pharmacological agents, especially diphenyl diselenide
The manuscript has interesting data as it compares the two analogue compounds, however some minor issues must be answered to improve the quality of the paper:
1) In introduction or methodology section, please mention the LD50 for these compounds in mice;
2) it is described that diphenyl diselenide also causes damage to mice, especially when administered more than once. Have authors treated these mice longer than in this protocol? In therapeuthics, usually people get more than one dose of some medicine. Hence, I would suggest authors to treat mice longer and compare the diphenyl diselenide mice. If authors are not willing to test that, please explain in the manuscript why this evaluation would not be necessary;
3) Have authors tried a dose response curve for diphenyl ditelluride? That would be interesting data to add, in order to observe at which concentration the DNA damage starts.
4) Please verify English language, there are few mistakes that can be corrected by the authors

Experimental design

Experimental design is well designed, however this reviewer thinks that further experiments evaluating chronic administration and dose-response curve could be a great add to this manuscript

Validity of the findings

Findings are relevant as authors mention that the doses can be therapeutically used, hence, toxicological observations need to be performed. DNA damage can cause irreversible problems to the cell and needs to be investigated for new drugs.

Additional comments

The manuscript brings interesting data, however I suggest, at least, a dose-response curve for diphenyl ditelluride.

Reviewer 3 ·

Basic reporting

In general, this manuscript is technically sound and the conclusion is made from reasonable experimental observations.

Experimental design

Reasonable experiments are design for major conclusion. However, more doses of the compounds are suggested to show the dose range for a toxic/protective effect. Otherwise the dose used in the manuscript was not in the pathological relevant range.

Validity of the findings

Proper controls are used, and the conclusion is made from proper analysis of the data.

Additional comments

More doses, especially lower doses are highly suggested to show the range for a toxic/protective effect of these two compounds. If 500umol/kg is the only (or lowest) dose to induce genotoxicity effect for (PhSe)2, then a single injection of 11 grams of (PhSe)2 will be need to have genotoxicity effect for a 70kg human. Thus, the major concern of this manuscript is that the dose of (PhSe)2 was not in a pathological relevant range.
In addition, if the observation that (PhSe)2 treatment induces free radical generation was well established as suggested by the authors, it was expected to see genetoxicity effect at this high dose. Thus, the novelty of this manuscript is greatly limited.

---

## Round 0.3 · accepted · Accept

The revised manuscript entitled “Differential Genotoxicity of Diphenyl Diselenide (PhSe)2 and Diphenyl Ditelluride (PhTe)2” has been greatly improved, and authors has successfully addressed all questions and concerns from 4 reviewers. In general, this study addressed an important question of differential genotoxic potential of (PhSe)2 and (PhTe)2 in mice. This study is also a logical extension of author’s contributions in the field. The manuscript is technically sound and the conclusion is made from reasonable experimental design. The revised manuscript title, the added references and improved data presentation all improved this manuscript and it is now in a form acceptable for publication.